# The Differential Effects of Internal Control Teams on Investment Decision Making Based on Industry Competition

Hyunjung Choi

College of Management and Technology, Sungkyul University, Anyang-si 14097, Republic of Korea;
hjchoi@sungkyul.ac.kr

**Abstract:** This study investigates how a company's internal control team affects their investment decision making, considering the level of industry competition within the South Korean capital market. A model obtained from the literature was employed to test the hypothesis. When industry competition is low, the quantitative adequacy of internal control staff increases the likelihood of investment when the risk of underinvestment is high, and it decreases the likelihood of investment when the risk of overinvestment is high. However, this is not the case when industry competition is fierce. Qualitative adequacy of internal control staff—expertise—has a significant effect on investment decision making when industry competition is high, but has no significant effect when industry competition is low. These results suggest that investors should consider the quantitative and qualitative adequacy of internal control staff along with the level of industry competition when evaluating the investment efficiency of a company.

**Keywords:** investment efficiency; industry competition; internal control team

## 1. Introduction

A company's investment decisions are an important outcome entrusted to managers. Proper or improper execution of investments directly affects business continuity (McConnell and Muscarella 1985; Klammer and Wilner 1991; Harris and Raviv 1996; Choi and Bae 2014). Therefore, companies should focus on efficient investment decision making. Internal control teams significantly impact corporate accounting, management decision-making, and financial reporting, and they play a role in improving the accuracy and reliability of disclosed information (Kim et al. 2012). Quantitative adequacy refers to the availability of sufficient staff in a firm, whereas qualitative adequacy refers to expertise. When internal control staff within companies are appropriately structured and the system operates efficiently, information asymmetry between managers and external stakeholders will be reduced. Further, effective monitoring and supervision of managers will enable them to make efficient investment decisions for companies (Choi 2023).

Managers must make strategic investments considering the level of competition within the industry. Companies experiencing high levels of intra-industry competition are more likely to undergo mergers and acquisitions compared to those that do not; this can occur through a change of managers (Holmstrom 1982; Kruse and Rennie 2006; Nalebuff and Stiglitz 1983). However, when industry competition is fierce, managers may overinvest to win the competition. Managers' compensation is often performance-based as it reduces agency problems between managers and shareholders. In this case, managers may overinvest if their performance compensation contract is asymmetric (Holmstrom and Milgrom 1991; Marino and Zabojnik 2008). Additionally, as industry competition increases, the possibility of disclosing company information to external stakeholders decreases, resulting in increased information asymmetry. This obstructs external stakeholders from efficiently monitoring, leading to managers overinvesting (Darrough and Stoughton 1990; Lanen and Verrecchia 1987). Particularly, in South Korea, the

higher the level of competition within an industry, the more excessive investment occurs (Cho and Choi 2016).

In situations in which managers make inefficient investment decisions when industry competition is high, it is necessary to examine the role of internal control team composition in investment decision making. It can be inferred that the positive relationship between an appropriate composition of internal control staff and a company's investment decision making may vary based on the level of industry competition. Therefore, this study examines how the relationship between employees in charge of internal control and investment efficiency varies according to the level of industry competition. Particularly, this study focuses on the adequacy of internal control staff from quantitative and qualitative perspectives. Since 2002, South Korea has been disclosing the departments and the number of employees in charge of internal control, including their credentials, making it the only country to do so. The South Korean government discloses this information to allow external stakeholders to evaluate the operation of internal controls when evaluating a company's value. Therefore, this study comprehensively examines the effect of the internal control workforce on investment efficiency according to the level of industry competition.

## 2. Prior Studies and Hypothesis Development

Competition within a company's industry affects both management's investment decisions and the effectiveness of its internal controls. Kim and Kim (2017) found that companies operating in competitive markets are more likely to have material weaknesses under SOX Section 404. They are also more likely to disclose multiple internal control weaknesses. The results show that market competition reduces the effectiveness of internal control over financial reporting and reduces the quality of a company's information environment. Moreover, the problems of weak internal control and the quality of accounting information are improved through quantitative and qualitative investment in internal control staff. Ryu et al. (2012) reported that the higher the number of internal control staff out of total employees and the longer the average number of years of experience of internal control staff, the lower the possibility of accounting errors. They explained that when a company has a sufficient number of internal control staff, effective internal control is achieved through appropriate division of work and timely review and monitoring of accounting functions. In addition, as internal control staff repeatedly perform internal control tasks, their understanding of the company deepens, the learning effect increases, and errors in accounting information are more effectively controlled. That is, the higher the competition within the industry, the more likely it is that an effective internal control system will be in place if internal control staff are well-invested.

The degree of competition within an industry can affect a company's investment efficiency. First, competition between companies within an industry can cause overinvestment in two main ways. In a highly competitive environment, if managers' incentives to build an empire increase, overinvestment may occur (Schumpeter 2021). Second, when competition is intense, the private cost of information disclosure increases, in which case the possibility of disclosing detailed information to external stakeholders decreases, which may lead to overinvestment (Lanen and Verrecchia 1987). However, there is also a possibility that the level of competition may suppress the possibility of overinvestment by managers and induce efficient investment. When competition is intense, managers are likely to make efficient investments because if they do not manage effectively, they may fall behind the competition and their job status could be at risk (Hart 1983). Contrastingly, if competition is intense and external stakeholders request more detailed information about the company, the possibility of overinvestment will decrease (Cheng et al. 2013). A similar logic applies to underinvestment. When competition is intense, the risk of bankruptcy is high, and managers are reluctant to make risky investments; additionally, when competition causes private costs of information disclosure, underinvestment may occur (La Porta et al. 2000). However, if competition forces managers to make their best efforts or increases the level of information disclosure, the level of competition may suppress underinvestment and

induce efficient investment (Bamber and Cheon 1998; Darrough and Stoughton 1990). That is, market competition can cause more overinvestment or underinvestment, or conversely, it can make investment levels more efficient. In their analysis of South Korean companies, Cho and Choi (2016) reported that as the level of competition within the industry increases, overinvestment—a negative role of competition—increases. Therefore, as competition within the industry intensifies, South Korean companies tend to overinvest and make inefficient investment decisions.

A significant relationship exists between the size of internal control departments and management's investment decisions. When the number of employees for internal control is sufficient, managers can be effectively monitored, and the quality of accounting information can be improved. Companies with a sufficient internal control workforce make more efficient investment decisions than those without one (Choi 2023). Summarizing previous studies, maintaining appropriate staff levels for internal control reduces information asymmetry and improves the quality of accounting information, thereby mitigating overinvestment owing to industry competition. If internal control staff enable managers to prioritize ethical decision making for the company, rather than pursuing personal gain, and mitigate opportunistic behavior and agency problems, a well-structured internal control team will enable efficient investment decision making, even when industry competition is high.

Both quantitative and qualitative adequacy should be considered when investigating the relationship between internal control teams and efficient investment decision making. Internal control departments may not be effective because of a lack of staff (Ge and McVay 2005). Furthermore, the expertise of staff must be ensured (Lee et al. 2012). When internal control staff handling accounting lack expertise, it becomes challenging to monitor managers and produce high-quality accounting information. Therefore, this study establishes the following hypothesis by categorizing internal control staff adequacy into quantitative and qualitative aspects to examine how their relationship with investment efficiency varies according to the level of industrial competition.

**Hypothesis:** *The relationship between the adequacy of internal control staff and investment efficiency varies based on the level of industry competition.*

### 3. Research Methods and Model

*3.1. Sample Selection*

The sample includes South Korean companies listed on the stock exchange. Companies from which internal control data could be obtained through annual business reports disclosed in the electronic disclosure system of the Financial Supervisory Service in South Korea from 2011 to 2018 were selected; data were collected manually. Additionally, among non-financial businesses, only corporations whose settlements were made in December were targeted. Financial data were obtained from KIS-VALUE and governance data were obtained from TS-2000.

*3.2. Measurement of Variables*

3.2.1. Quantitative Adequacy of Internal Control Teams

The quantitative adequacy of internal control teams is measured using Equation (1), following Lee et al. (2012).

$$\ln(\text{IC}) = \alpha + \beta_1 SIZE + \beta_2 \ln(AFFIL) + \beta_3 EXPT + \beta_4 \ln(EMPL) + \beta_5 ROA$$
$$+ \beta_6 LEV + \sum YR\ Dummy + \epsilon \tag{1}$$

If the residual value of Equation (1) is greater than 0, the size of internal control staff is considered adequate when compared to other companies with the same conditions; $IC\_H$ has a value of 1 or is 0 otherwise. In the above equation, the dependent variable is the natural logarithm of the number of staff for internal control. The independent variable is a



characteristic variable representing the complexity of a company's accounting processes (Lee et al. 2012). Appendix A presents the details of these variables.

### 3.2.2. Qualitative Adequacy of Internal Control Teams

The qualitative adequacy of internal control teams refers to their credentials and is measured by the possession of sufficient experience and whether they are certified public accountants (hereafter, CPA). To assess the sufficiency of experience, a dummy variable with a value of 0 is assigned if *EXP_H* exceeds the median based on the average months of experience in the internal control department. Alternatively, it is measured by the presence or absence of CPAs within the internal control team. If at least one CPA is present, *CPA_D* is set to 1; otherwise, a dummy variable with a value of 0 is assigned.

### 3.2.3. Investment Efficiency

To measure investment efficiency, this study assumes a prior situation in which there is a high risk of overinvestment or underinvestment based on the size of a company's cash holdings and debt ratio (Biddle et al. 2009). When companies finance investments, internally held cash is used if cash flow from operating activities is insufficient; otherwise, debt is used. However, if cash holdings are low and the debt ratio is high, the risk of bankruptcy increases with additional financing, thus making it difficult to secure investment funds.

The risk of overinvestment (*OVER*) is calculated as follows based on the research method of Cho and Choi (2016). First, cash and cash equivalents are divided by total assets and the value of the 10th quantile of the entire sample is derived. Additionally, the value obtained by multiplying the debt ratio by $-1$ derives the value of the 10th quantile of the entire sample. After adjusting each value between 0 and 1, the average of the two values is calculated. When the risk of overinvestment is high, the value of *OVER* is close to 1, and it is close to 0 when the risk of underinvestment is high.

### 3.2.4. Industrial Competition

To measure the level of competition within an industry, the Herfindahl–Hirschman Index (*HHI*), which has been used in several previous studies, is used (Dhaliwal et al. 2014). After deriving the market share, *HHI* is calculated as the sum of the squares of the market share (Grullon et al. 2019; Park and Kwon 2012). Market share is calculated as the percentage of sales of individual companies out of the total sales of companies belonging to each industry. The higher the *HHI*, the lower the level of industry competition, and the lower the *HHI*, the higher the level of industry competition.

### 3.3. Research Model

To test the study hypothesis, Equation (2) is used. The sample is categorized based on the level of industry competition—high and low—and verified. Based on the median of the *HHI*, if the *HHI* value is higher than the median, it is classified as a group with a low degree of industry competition, and if it is lower than the median, it is classified as a group with a high degree of industry competition.

$$INVESTMNET_{t+1}$$
$$= \beta_0 + \beta_1 IC\_AD_t + \beta_2 IC\_AD_t \times OVER_t + \beta_3 OVER_t + \beta_4 SIZE$$
$$+ \beta_5 LEV_t + \beta_6 ROA_t + \beta_7 MB_t + \beta_8 PPEA_t + \beta_9 STDCFO_t \qquad (2)$$
$$+ \beta_{10} STDSALES_t + \beta_{11} STDINVESTMENT_t$$
$$+ \beta_{12} MOWN_t + \beta_{13} FOWN_t + \sum IND + \sum YR + \varepsilon_1$$

In Equation (2), the dependent variable *INVESTMENT*, refers to the subsequent investment amount, calculated by subtracting cash inflow from the disposal of tangible assets from the sum of capital investment and R&D expenses (Cho and Choi 2016). The adequacy

of internal control teams (*IC_AD*), a variable of interest for verifying the hypothesis, is verified by categorizing it into quantitative adequacy (*IC_H*) and qualitative adequacy (*EXP_H, CPA_D*). In Equation (2), $\beta_1$ is a situation in which the company is more likely to underinvest and represents the effect of the adequacy of staff on investments in a situation of "*OVER* = 0". In a situation with a high risk of underinvestment, if the investments of a company with adequate staff for internal control increase, $\beta_1$ will show a significantly positive value. $\beta_2$ represents an additional relationship between an adequately sized internal control team and investments when the company's risk of overinvestment is high. $\beta_1 + \beta_2$ is a situation in which *OVER* = 1, which indicates the correlation between having adequate staff for internal control and investments when there is a tendency to overinvest. In a situation in which there is high propensity to overinvest, if investments decrease when the composition of the internal control team is appropriate, $\beta_1 + \beta_2$ has a significant negative value. This means that companies do not overinvest when there is a high risk of overinvestment; in this case, investment efficiency is considered high. According to previous studies, variables that are expected to affect investments are company size (*SIZE*), debt ratio (*LEV*), return on assets (*ROA*), growth potential (*MB*), tangible assets subject to depreciation (*PPEA*), operating cash flow volatility (*STDFO*), sales volatility (*SRDSALES*), investment volatility (*STDINVESTMENT*), management equity ratio (*MOWN*), and foreign shareholder equity ratio (*FOWN*); these are included in Equation (2) (Biddle et al. 2009; Myers 1977; Richardson 2006). Appendix A provides detailed descriptions of these variables.

## 4. Empirical Results and Discussion

### 4.1. Descriptive Statistics

Table 1 presents the descriptive statistics of the variables. Winsorization was performed at the upper and lower one percent levels for all continuous variables. The average of *INVESTMENT* is 4.525, and the average of the annual investment amount of the sample companies is approximately 4.5% of their assets at the beginning of the year. A decently sized sample of internal control staff is 45.3%. Regarding the credentials of internal control staff, the average amount of work experience is 122.166 months, and 18.8% of companies have employees who are CPAs. The mean of *HHI* is 0.124, and the median is 0.094. *OVER* has an average of 0.563 and a median of 0.550.

**Table 1.** Descriptive Statistics.

| Variables | Mean | Min | Median | Max | Std. Dev. |
|---|---|---|---|---|---|
| *INVESTMENT* | 4.525 | −3.791 | 3.043 | 23.771 | 4.847 |
| *IC_H* | 0.453 | 0.000 | 0.000 | 1.000 | 0.498 |
| *EXP* (months) | 122.166 | 0.000 | 103.444 | 2035.000 | 118.580 |
| *CPA_D* | 0.188 | 0.000 | 0.000 | 1.000 | 0.391 |
| *HHI* | 0.124 | 0.052 | 0.094 | 0.427 | 0.083 |
| *OVER* | 0.563 | 0.100 | 0.550 | 1.000 | 0.391 |
| *SIZE* | 27.147 | 24.562 | 26.870 | 31.297 | 0.218 |
| *LEV* | 0.427 | 0.050 | 0.428 | 0.927 | 1.466 |
| *ROA* | 0.025 | −0.241 | 0.028 | 0.187 | 0.199 |
| *MB* | 1.314 | 0.227 | 0.937 | 7.870 | 0.062 |
| *PPEA* | 0.194 | 0.004 | 0.170 | 0.609 | 1.255 |
| *STDCFOA* | 0.049 | 0.001 | 0.041 | 1.137 | 0.132 |
| *STDSALES* | 0.129 | 0.003 | 0.096 | 1.777 | 0.044 |
| *STDINVESTMENT* | 3.379 | 0.017 | 2.179 | 51.997 | 0.129 |
| *MOWN* | 0.440 | 0.107 | 0.439 | 0.808 | 4.103 |
| *FOWN* | 0.111 | 0.000 | 0.054 | 0.606 | 0.158 |

Variables are defined in Appendix A.

Correlation coefficients between variables are not presented; however, *INVESTMENT* shows a significant negative correlation with *IC_H*, indicating that companies with a well-structured internal control personnel team have high capital expenditures. Additionally, the

correlation coefficient between *INVESTMENT* and *OVER* is 0.061, showing a significantly positive value, suggesting that the likelihood of investments is high when the risk of overinvestment is high. Regarding the correlation between *INVESTMENT* and the control variables, *SIZE, ROA, MB,* and *PPEA* show significant positive correlation coefficients, but it has a significant negative correlation coefficient with *LEV*.

*4.2. Hypothesis Results*

Table 2 presents the results of the hypothesis testing. Column (1) presents the analysis of the entire sample, column (2) presents the sample with high industry competition, and column (3) presents the verification of Equation (2) for the sample with low industry competition.

**Table 2.** Quantitative Adequacy of Internal Control Teams and Investment Decision Making.

| Variables | Dependent Variable: *INVESTMENT* | | |
|---|---|---|---|
| | **Total Sample** | **Sample with High Industry Competition** | **Sample with Low Industry Competition** |
| *Intercept* | 0.695 (0.36) | 0.623 (1.91) * | 1.206 (0.51) |
| *IC_H* ($\beta_1$) | 0.678 (3.54) *** | 0.665 (1.21) | 0.685 (2.70) *** |
| *IC_H × OVER* ($\beta_2$) | −2.788 (−3.17) *** | −2.188 (−1.64) | −3.125 (−2.58) *** |
| $\beta_1 + \beta_2$ (*F-value*) | −2.110 (7.16) *** | −1.523 (1.67) | −2.440 (4.94) ** |
| *OVER* | 1.442 (2.22) ** | 1.790 (1.75) * | 0.877 (1.02) |
| *SIZE* | 0.056 (0.84) | 0.270 (2.19) ** | 0.081 (0.99) |
| *LEV* | 0.262 (0.40) | −1.657 (−1.56) | −0.933 (−1.06) |
| *ROA* | 1.661 (11.87) *** | 1.506 (6.51) *** | 1.762 (9.87) *** |
| *MB* | 0.659 (9.86) *** | 0.650 (7.13) *** | 0.561 (5.40) *** |
| *PPEA* | 0.904 (13.64) *** | 0.827 (8.13) *** | 0.981 (10.95) *** |
| *STDCFOA* | 0.824 (0.45) | 0.338 (0.09) | 1.503 (0.70) |
| *STDSALES* | −1.249 (−2.03) ** | −1.230 (−1.94) * | −0.873 (−1.19) |
| *STDINVESTMENT* | 0.079 (3.93) *** | 0.089 (2.83) *** | 0.100 (3.66) *** |
| *MOWN* | 0.295 (0.58) | 1.857 (2.33) ** | 1.935 (2.80) *** |
| *FOWN* | 0.990 (1.29) | 0.824 (0.60) | 1.641 (2.78) *** |
| *Industry Dummy* | Included | Included | Included |
| *Year Dummy* | Included | Included | Included |
| Adj $R^2$ | 0.23 | 0.22 | 0.22 |
| F | 36.24 *** | 19.60 *** | 25.83 *** |
| n | 3945 | 1762 | 2183 |

Variables are defined in Appendix A. t-statistics are reported in parentheses. * $p < 0.1$, ** $p < 0.05$, *** $p < 0.01$ (two-tailed tests).

Column (1) includes the revalidation of previous studies targeting the entire sample. The coefficient of IC_H is 0.678, and the t-value is 3.54, which is significant ($p < 0.01$). The coefficient of IC_H × OVER is −2.278, and the t-value is 3.17, which is significant ($p < 0.01$). Furthermore, $\beta_1 + \beta_2$ is −2.110, which is significant ($p < 0.01$). If the amount of staff for internal control is adequate, investments increase when the risk of underinvestment is higher than in other cases, and decrease when the risk of overinvestment is high, ultimately leading to efficient investment decisions.

The results of the hypothesis testing are presented in columns (1) and (2). In column (2), neither the coefficient for IC_H nor the coefficient of IC_H × OVER are significant. If the level of industry competition is high and staff for internal control are adequate, it indicates that companies are unlikely to invest when there is a high risk of underinvestment or overinvest when the risk of overinvestment is high. Column (3) presents a case of low industry competition; the coefficient of IC_H is 0.685, and the t-value is 2.70, which is significant ($p < 0.01$). The coefficient of IC_H × OVER is −3.125, and the t-value is 2.58, which is significant ($p < 0.01$). Furthermore, $\beta_1 + \beta_2$ is significant ($p < 0.01$) at −2.440. Investments increase when the risk of underinvestment is high and decrease when the risk of overinvestment is high. Combining these results, the effect of the adequacy

of internal control teams on investment decision making differs based on the level of industry competition.

　　Table 3 shows the results of the hypothesis testing relating to the qualitative adequacy of staff. Panel A measures personnel expertise based on the months of employee experience and Panel B measures whether any of the employees are CPAs.

**Table 3.** Qualitative Adequacy of Internal Control Teams and Investment Decision Making.

| Panel A: Experience | | | |
|---|---|---|---|
| | Dependent Variable: *INVESTMENT* | | |
| **Variables** | **Total Sample** | **Sample with High Industry Competition** | **Sample with Low Industry Competition** |
| *Intercept* | 0.036 (0.02) | 0.738 (3.14) *** | 1.394 (0.82) |
| *EXP_H ($\beta_1$)* | 0.261 (0.62) | 0.909 (1.92) * | −0.473 (−1.24) |
| *EXP_H × OVER ($\beta_2$)* | 0.133 (0.19) | −1.391 (−1.85) * | −0.141 (−0.22) |
| *$\beta_1 + \beta_2$ (F-value)* | 0.394 (1.36) | −0.482 (3.91) ** | −0.614 (2.36) |
| *OVER* | 0.795 (1.98) ** | 2.168 (3.09) *** | 0.557 (1.96) ** |
| *Controls* | Included | Included | Included |
| Adj R$^2$ | 0.23 | 0.27 | 0.29 |
| F | 31.54 *** | 21.83 *** | 26.30 *** |
| n | 3945 | 1762 | 2183 |
| **Panel B: CPA** | | | |
| | Dependent Variable: *INVESTMENT* | | |
| **Variables** | **Total Sample** | **Sample with High Industry Competition** | **Sample with Low Industry Competition** |
| *Intercept* | 1.135 (0.56) | 1.073 (4.60) *** | 1.521 (0.88) |
| *CPA_D ($\beta_1$)* | 0.517 (0.86) | 1.138 (1.88) * | −0.142 (−0.24) |
| *CPA_D × OVER ($\beta_2$)* | −0.317 (−0.32) | −0.272 (−0.29) | 0.455 (0.46) |
| *$\beta_1 + \beta_2$ (F-value)* | 0.200 (0.17) | 0.866 (3.89) ** | 0.313 (0.38) |
| *OVER* | 0.295 (2.50) ** | 1.577 (2.56) ** | 0.301 (1.97) ** |
| *Controls* | Included | Included | Included |
| Adj R$^2$ | 0.23 | 0.28 | 0.28 |
| F | 35.98 *** | 22.28 *** | 24.58 *** |
| n | 3945 | 1762 | 2183 |

Variables are defined in Appendix A. t-statistics are reported in parentheses. * $p < 0.1$, ** $p < 0.05$, *** $p < 0.01$ (two-tailed tests).

　　In Panel A, the coefficients for both EXP_H and EXP_H × OVER in column (1) are non-significant. When the internal control workforce possesses adequate credentials, the decision is made to not reduce investments when the risk of underinvestment is high or increase investments when the risk of overinvestment is high. In column (2), the coefficients of EXP_H and EXP_H × OVER are 0.909 and −1.391, respectively, which are significant ($p < 0.1$). If the expertise of the staff for internal control is adequate, investments increase when the risk of underinvestment is high and decrease when the risk of overinvestment is high, resulting in efficient investment decision making. However, neither the EXP_H nor the EXP_H × OVER coefficients are significant in column (3), similar to column (1).

　　As shown in Panel B, the coefficients of both CPA_D and CPA_D × OVER in column (1) are non-significant. However, the coefficient of CPA_D in column (2) is significant ($p < 0.1$) at 1.138, suggesting that if internal control staff are CPAs, they do not underinvest when there is a risk of underinvestment, and investments are increased. However, the coefficient of CPA_D × OVER is non-significant, suggesting that when the risk of overinvestment is high, whether an employee is a CPA has no additional effect on investments. In column (3), the coefficients of both CPA_D and CPA_D × OVER are non-significant, indicating that the possession of a CPA license does not have a significant additional effect on investment decision making when the level of industry competition is low. In summary,

when competition within the industry is low, the expertise of internal control staff does not have a significant effect on investment decisions. When competition within an industry is high, the expertise of internal control staff has a significant impact on investment decisions. Although there are some differences depending on the type of expertise of the employee in charge, it is possible to make efficient investment decisions when the level of industry competition is high.

### 4.3. Sensitivities

To strengthen the results, sensitivities were conducted. First, the results described above were verified using the value (OVER) calculated by dividing the 10 quantiles based on the level of cash holdings and debt ratio, which are variables indicating the risk of overinvestment. Additionally, the hypothesis was retested using continuous variables instead of deciles. The verification results were identical to those listed in Tables 2 and 3. Second, the hypothesis was reexamined, following Moon et al. (2012) in addition to the method suggested by Lee et al. (2012), as a variable to measure the quantitative adequacy of staff. The methodology proposed by Moon et al. (2012) uses the ratio of the number of internal control personnel to the total number of employees. The verification results were identical to those in Table 2.

## 5. Conclusions

This study investigated the effects of the quantitative and qualitative adequacy of internal control teams on investment decision making according to the degree of industry competition. The results show that the effect of the adequacy of the internal control workforce on investment efficiency differs based on the level of industry competition. During high levels of industry competition, and when the internal control workforce is adequate, investments are not reduced when the risk of underinvestment is high nor increased when the risk of overinvestment is high. Contrastingly, when the level of industry competition is low, investments are increased when the risk of underinvestment is high and decreased when the risk of overinvestment is high. The expertise of internal control personnel has a significant effect on investment decision making when the level of industry competition is high, but no significant effect is seen when the level of industry competition is low. When industry competition is low, the quantitative adequacy of internal control personnel has a significant impact on investment efficiency; however, when industry competition is high, the expertise of internal control personnel has a significant impact on investment efficiency. The quantitative or qualitative adequacy of internal control personnel is a mechanism for effective investment decision making, which means that the internal control team acts differently depending on the level of industry competition which the company faces.

This study has the following implications. First, the results suggest that when evaluating a company's investment efficiency from an investor's perspective, the number of internal control employees in organizations and their credentials should be considered, along with the level of industry competitiveness. Second, from the perspective of a company, this study can be a basis for raising awareness on human resource investment in internal control and making investments. It is an opportunity for companies to increase their corporate value by establishing systems, such as setting a minimum number of internal control personnel. Finally, this study provides an opportunity to expand research on the effectiveness of the internal control workforce. Most existing internal control studies focus on whether there are vulnerabilities. However, the primary reason for weak internal control or poor quality of accounting information is because the composition of internal control is inadequate in terms of quantity and quality. The current results broaden the scope of related research in the future.

**Funding:** This research received no external funding.

**Informed Consent Statement:** Not applicable.

**Data Availability Statement:** Internal control data can be obtained through the annual business reports disclosed in the electronic disclosure system of the Financial Supervisory Service in Korea. This data can be found here: https://dart.fss.or.kr (accessed on 31 December 2019). Financial data were obtained from KIS-VALUE and governance data were obtained from TS-2000.

**Conflicts of Interest:** The author declares no conflict of interest.

## Appendix A

| Variable | Definition |
| --- | --- |
| INVESTMENT | (Capital investment + R&D − cash inflow from disposal of tangible assets)/total assets for term t − 1 |
| IC_H | 1 if the number of employees in the internal control department is adequate and 0 otherwise |
| EXP_H | 1 if the average experience of employees in the internal control department is higher than the median and 0 otherwise |
| CPA_D | 1 if there is a certified public accountant among employees in the internal control department and 0 otherwise |
| OVER | Average of cash and debt ratios divided into 10 quantiles |
| SIZE | Natural logarithm of total assets |
| LEV | Total liabilities/total assets |
| ROA | Net income/total assets |
| MB | Market value/equity |
| PPEA | (Tangible assets−land−assets under construction)/total assets for term t − 1 |
| STDCFO | Standard deviation of cash flow from operating activities/total assets for term t − 1 over the past five years |
| STDSALES | Standard deviation of sales/average assets over the past five years |
| STDINVESTMENT | Standard deviation of investments over the past five years |
| MOWN | Owners' shareholder ratio |
| FOWN | Foreigners' shareholder ratio |
| ln (IC) | Natural logarithm of the number of employees in the internal control department |
| ln(AFFIL) | Natural logarithm of the number of associates |
| EXPT | Share of exports in total sales |
| ln(EMPL) | Natural logarithm of the total number of employees |

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
