# Peer review of "The Differential Effects of Internal Control Teams on Investment Decision Making Based on Industry Competition"

_ijfs, doi:10.3390/ijfs11040131_

Round 1
Reviewer 1 Report
Comments and Suggestions for Authors
The topic of this article is of interest to wide readers. The paper's argument is built on an appropriate base of theory and the methods employed in the paper are appropriate. The literature review is overly descriptive and would benefit from greater critical analysis of the extant research in the field. The literature review is not very organized and at times is not up to date. The authors should have cited more recent papers. The study requires considerable attention concerning writing style, grammar, and punctuation.
Comments on the Quality of English LanguageThe study requires considerable attention concerning writing style, grammar, and punctuation.
Author Response
Thank you for pointing this out. I agree with this comment. Considering that market competition affects both management’s decisions and the effectiveness of internal controls, I revised to focus on how market competition relates to effective internal control and how market competition relates to investment decisions (pages 2–3, lines 64–128): “Competition within a company’s industry affects both management’s investment decisions and the effectiveness of its internal controls. Kim and Kim (2014) found that companies operating in competitive markets are more likely to have material weaknesses under SOX Section 404. They are also more likely to disclose multiple internal control weaknesses. The results show that market competition reduces the effectiveness of internal control over financial reporting and reduces the quality of a company's information environment. Moreover, the problems of weak internal control and the quality of accounting information are improved through quantitative and qualitative investment in internal control staff. Ryu et al. (2012) reported that the higher the number of internal control staff in total employees and the longer the average number of years of experience of internal control staff, the lower the possibility of accounting errors. They explained that when a company has a sufficient number of internal control staff, effective internal control is achieved through appropriate division of work and timely review and monitoring of accounting functions. In addition, as internal control staff repeatedly perform internal control tasks, their understanding of the company deepens, the learning effect increases, and errors in accounting information are more effectively controlled. That is, the higher the competition within the industry, the more likely it is that an effective internal control system will be in place if internal control staff are well invested.
The degree of competition within an industry can affect a company's investment efficiency. First, competition between companies within an industry can cause overinvestment in two main ways. In a highly competitive environment, if managers’ incentives to build an empire increase, overinvestment may occur (Schumpeter 2021). Second, when competition is intense, the private cost of information disclosure increases, in which case the possibility of disclosing detailed information to external stakeholders decreases, which may lead to overinvestment (Lanen and Verrecchia 1987). However, there is also a possibility that the level of competition may suppress the possibility of overinvestment by managers and induce efficient investment. When competition is intense, managers are likely to make efficient investments because if they do not manage effectively, they may fall behind the competition and their job status could be at risk (Hart 1983). Contrastingly, if competition is intense and external stakeholders request more detailed information about the company, the possibility of overinvestment will decrease (Cheng et al. 2013). A similar logic applies to underinvestment. When competition is intense, the risk of bankruptcy is high, and managers are reluctant to make risky investments; or when competition causes private costs of information disclosure, underinvestment may occur (La Porta et al. 2000). However, if competition forces managers to make their best efforts or increases the level of information disclosure, the level of competition may suppress underinvestment and induce efficient investment (Bamber and Cheon 1998; Darrough and Stoughton 1990). That is, market competition can cause more overinvestment or underinvestment, or conversely, it can make investment levels more efficient. In their analysis of South Korean companies, Cho and Choi (2016) reported that as the level of competition within the industry increases, overinvestment—a negative role of competition—increases. Therefore, as competition within the industry intensifies, South Korean companies tend to overinvest and make inefficient investment decisions.
A significant relationship exists between the size of internal control departments and management’s investment decisions. When the number of employees for internal control is sufficient, managers can be effectively monitored, and the quality of accounting information can be improved. Companies with a sufficient internal control workforce make more efficient investment decisions than those without one (Choi 2023). Summarizing previous studies, maintaining appropriate staff levels for internal control reduces information asymmetry and improves the quality of accounting information, thereby mitigating overinvestment owing to industry competition. If internal control staff enable managers to prioritize ethical decision-making for the company, rather than pursuing personal gain and mitigate opportunistic behavior and agency problems, a well-structured internal control team will enable efficient investment decision-making, even when industry competition is high.
Both quantitative and qualitative adequacy should be considered when investigating the relationship between internal control teams and efficient investment decision-making. Internal control departments may not be effective because of a lack of staff (Ge and McVay 2005). Further, the expertise of staff must be ensured (Lee et al. 2012). When internal control staff handling accounting lack expertise, it becomes challenging to monitor managers and produce high-quality accounting information. Therefore, this study establishes the following hypothesis by categorizing internal control staff adequacy into quantitative and qualitative aspects to examine how their relationship with investment efficiency varies according to the level of industrial competition.”

Reviewer 2 Report
Comments and Suggestions for Authors
This paper investigates how the number of staff in an internal control team and the experiences of the staff affect the managerial investment decision when industry competition is either high or low. Investment decision is efficient when investments are not reduced if the risk of underinvestment is high nor increased if the risk of overinvestment is high. The author finds that there is a positive association between the internal control adequacy and efficient investment decision when the internal control adequacy is measured by the number of staff.
(1) Market competition affects both managerial decision and the effectiveness of internal control. For example, Kim and Kim (2014 Asia-Pacific Journal of Accounting and Economics) demonstrate that firms operating in competitive markets are more likely to disclose internal control weaknesses. Please provide some literature reviews on (i) how market competition is related to effective internal control; and (ii) how market competition is related to investment decision.
(2) Your results for the high competition subsample are weak. I wonder whether this can be explained as high market competition is an alternative mechanism for effective internal control.
Comments on the Quality of English LanguageThe paper is well written. Some sentences are a bit too long (for example, see lines 93-96). I suggest some minor edit.
Author Response
|
Comments 1: Market competition affects both managerial decision and the effectiveness of internal control. For example, Kim and Kim (2014 Asia-Pacific Journal of Accounting and Economics) demonstrate that firms operating in competitive markets are more likely to disclose internal control weaknesses. Please provide some literature reviews on (i) how market competition is related to effective internal control; and (ii) how market competition is related to investment decision. |
|
Response 1: Thank you for pointing this out. I agree with this comment. The points raised by the reviewer: (i) how market competition is related to effective internal control, and (ii) how market competition is related to investment decision are described as follows. I revised the “Prior Studies and Hypothesis Development” section of the paper to reflect these aspects (please see page 2, lines 64–81). “Kim and Kim (2014) found that companies operating in competitive markets are more likely to have material weaknesses under SOX Section 404. They are also more likely to disclose multiple internal control weaknesses. The results show that market competition reduces the effectiveness of internal control over financial reporting and reduces the quality of a company's information environment. Moreover, the problems of weak internal control and the quality of accounting information are improved through quantitative and qualitative investment in internal control personnel. Ryu et al. (2012) reported that the higher the number of internal control staff in total employees and the longer the average number of years of experience of internal control staff, the lower the possibility of accounting errors. They explained that when a company has a sufficient number of internal control staff, effective internal control is achieved through appropriate division of work and timely review and monitoring of accounting functions. In addition, as internal control staff repeatedly perform internal control tasks, their understanding of the company deepens, the learning effect increases, and errors in accounting information are more effectively controlled. That is, the higher the competition within the industry, the more likely it is that an effective internal control system will be in place if internal control staff are well invested.” The degree of competition within an industry can affect a company's investment efficiency. First, competition between companies within an industry can cause overinvestment in two main ways. In a highly competitive environment, if managers’ incentives to build an empire increase, overinvestment may occur (Schumpeter 2021). Second, when competition is intense, the private cost of information disclosure increases, in which case the possibility of disclosing detailed information to external stakeholders decreases, which may lead to overinvestment (Lanen and Verrecchia 1987). However, there is also a possibility that the level of competition may suppress the possibility of overinvestment by managers and induce efficient investment. When competition is intense, managers are likely to make efficient investments because if they do not manage effectively, they may fall behind the competition and their job status could be at risk (Hart 1983). Contrastingly, if competition is intense and external stakeholders request more detailed information about the company, the possibility of overinvestment will decrease (Alchian 1950; Stigler 1958; Cheng et al. 2013). A similar logic applies to underinvestment. When competition is intense, the risk of bankruptcy is high, and managers are reluctant to make risky investments; or when competition causes private costs of information disclosure, underinvestment may occur (La Porta et al. 2000). However, if competition forces managers to make their best efforts or increases the level of information disclosure, the level of competition may suppress underinvestment and induce efficient investment (Bamber and Cheon 1998; Darrough and Stoughton 1990). That is, market competition can cause more overinvestment or underinvestment, or conversely, it can make investment levels more efficient. In their analysis of South Korean companies, Cho and Choi (2016) reported that as the level of competition within the industry increases, overinvestment—a negative role of competition—increases. Therefore, as competition within the industry intensifies, South Korean companies tend to overinvest and make inefficient investment decisions. |
|
Comments 2: Your results for the high competition subsample are weak. I wonder whether this can be explained as high market competition is an alternative mechanism for effective internal control. |
|
Response 2: Thank you for pointing this out. The results for the high competition subsample are indeed weak. Table 3 analyzes the relationship between the qualitative adequacy of internal control personnel and investment decision-making. The qualitative adequacy of internal control personnel was measured by sufficient experience and possession of certified public accountants. In case of sufficiency of experience, the median of the average number of months of experience by year was used as the standard. If it was higher than the median, the person-in-charge was classified as having sufficient experience. However, when measuring the presence of certified public accountants among internal control personnel, the number of cases where certified public accountants have been hired is still minimal. Looking at Table 1, companies with certified public accountants account for approximately 18% of the total sample. Ryu et al. (2012), a related previous study, also noted the same problem. However, having a certified public accountant is an important factor in determining the expertise of internal control. Accordingly, this study tested the hypothesis by considering two factors to determine the expertise of internal control personnel: sufficiency of experience and possession of a certified public accountant. I do not suggest that highly competitive markets are an alternative mechanism for effective internal controls. I suggest that when competition is fierce, the quantitative or qualitative adequacy of internal control personnel is a mechanism for effective investment decision-making. My intention is to suggest that as competition within the industry becomes more intense, not only quantitative but also qualitative investment in internal control personnel must be considered to make effective investment decisions from the perspective of companies and investors. In consideration of your helpful review, I revised the hypotheses and results sections. |
